# The Application of Radiomics and AI to Molecular Imaging for Prostate Cancer

**DOI:** 10.3390/jpm14030287

**Published:** 2024-03-07

**Authors:** William Tapper, Gustavo Carneiro, Christos Mikropoulos, Spencer A. Thomas, Philip M. Evans, Stergios Boussios

**Affiliations:** 1Centre for Vision Speech and Signal Processing, The University of Surrey, 388 Stag Hill, Surrey, Guildford GU2 7XH, UK; w.tapper@surrey.ac.uk (W.T.); g.carneiro@surrey.ac.uk (G.C.); p.evans@surrey.ac.uk (P.M.E.); 2National Physical Laboratory, Hampton Road, Teddington TW11 0LW, UK; spencer.thomas@npl.co.uk; 3Clinical Oncology, Royal Surrey NHS Foundation Trust, Egerton Road, Surrey, Guildford GU2 7XX, UK; christos.mikropoulos@nhs.net; 4Department of Medical Oncology, Medway NHS Foundation Trust, Gillingham ME7 5NY, UK; 5School of Cancer and Pharmaceutical Sciences, Faculty of Life Sciences and Medicine, King’s College London, Strand, London WC2R 2LS, UK; 6Kent and Medway Medical School, University of Kent, Canterbury CT2 7LX, UK; 7Faculty of Medicine, Health, and Social Care, Canterbury Christ Church University, Canterbury CT2 7PB, UK; 8AELIA Organisation, 9th km Thessaloniki–Thermi, 57001 Thessaloniki, Greece

**Keywords:** prostate cancer, molecular imaging, artificial intelligence, PET/CT, radiomics, machine learning

## Abstract

Molecular imaging is a key tool in the diagnosis and treatment of prostate cancer (PCa). Magnetic Resonance (MR) plays a major role in this respect with nuclear medicine imaging, particularly, Prostate-Specific Membrane Antigen-based, (PSMA-based) positron emission tomography with computed tomography (PET/CT) also playing a major role of rapidly increasing importance. Another key technology finding growing application across medicine and specifically in molecular imaging is the use of machine learning (ML) and artificial intelligence (AI). Several authoritative reviews are available of the role of MR-based molecular imaging with a sparsity of reviews of the role of PET/CT. This review will focus on the use of AI for molecular imaging for PCa. It will aim to achieve two goals: firstly, to give the reader an introduction to the AI technologies available, and secondly, to provide an overview of AI applied to PET/CT in PCa. The clinical applications include diagnosis, staging, target volume definition for treatment planning, outcome prediction and outcome monitoring. ML and AL techniques discussed include radiomics, convolutional neural networks (CNN), generative adversarial networks (GAN) and training methods: supervised, unsupervised and semi-supervised learning.

## 1. Introduction

Prostate cancer (PCa) is the second most frequent cancer diagnosis in men, and the fifth highest cause of death worldwide [1]. Early diagnosis of diseases is known to improve patient outcomes, and accurate subtyping and staging can inform better treatment plans and improved quality of life [2]. This suggests that research into improving how PCa is diagnosed and staged is highly important for patient survival by ensuring that PCa is not underdiagnosed, and molecular research is a key component of this. In addition, the rapidly growing application of artificial intelligence (AI) is transforming molecular imaging in PCa.

The scope for the use of AI in the PCa imaging pathway includes helping early detection and staging of disease, plus the prediction of outcome based on complex patterns in previous data. AI has the potential to support the oncologist in decision-making by providing extra analytical insight and also flagging features in unusual cases. In summary, AI has the potential to provide decision support tools and to support labor-saving.

The diagnosis of PCa relies on the microscopic examination of prostate tissue obtained through needle biopsy. A primary Gleason grade is assigned to the most prevalent histological pattern observed, while a secondary grade is assigned to the highest-grade pattern, based on the microscopic structure and appearance of the cells. Additionally, tests for serum prostate-specific antigen (PSA) variants can help estimate the likelihood of prostate cancer in patients who have previously undergone a negative biopsy. One such test is the PCa antigen 3 test, which involves analyzing urine collected after prostatic massage. This test has been validated in this patient population, demonstrating an 88% negative predictive value for subsequent biopsy [3]. Novel imaging technology has also been integrated into diagnostic pathways of PCa. A common approach is in Magnetic Resonance Imaging (MRI); however, there has been an increased use in Computed Tomography (CT) and Positron Emission Tomography (PET) in recent years since the application of the molecular biomarker Prostate Specific Membrane Antigen-PET (PSMA-PET) [4]. While MRI has been extensively covered in previous review articles [5,6], there is a lack of a detailed review of PET/CT within PCa. In addition, the role of both PET/CT and AI in prostate cancer imaging is rapidly growing. For these reasons, this article will review the use of PET/CT and AI.

This narrative review will provide an overview of imaging for PCa, with a focus on PET/CT and PSMA-PET, with comments on other modalities, such as MRI and ultrasound, and on histopathology where appropriate. The role of AI applied to imaging for lesion detection, staging, treatment planning and outcome prediction will be discussed.

This review will start with a summary of the use of PSMA for PET/CT. Then, the main body of the review will focus on the most commonly used AI methodologies applied to PCa imaging, including radiomics, convolutional neural networks (CNNs), the use of unsupervised and semi-supervised learning and generative adversarial networks (GANs) with applications to lesion detection, staging, treating planning and outcome prediction. This review will highlight the differences between supervised, semi-supervised, weakly supervised and unsupervised learning and their applications to PCa. Each section will begin with an explanation of the technology, how it functions and will then continue on to its applications.

## 2. PSMA-PET/CT as the Preferred Imaging Modality for Identifying Prostate Cancer Spread

PSMA-PET offers an innovative approach to accurately staging PCa. A multi-center randomized trial comparing conventional imaging to Gallium (Ga)-PSMA-PET showed its superiority in identifying pelvic nodes accurately (92% vs. 65% with conventional imaging). Increasingly, clinicians are using PSMA-PET, especially in patients with high-risk disease [4].

Another very important application of PSMA-PET is the detection of distant lesions and clarification of indeterminate lesions on conventional imaging in disease relapse. In the case of biochemical failure, PSMA-PET changes the intended treatment and allows for a more personalized approach in nearly 50% of cases, as quantified in a single arm study in post-prostatectomy patients [7].

The European Association of Urology (EAU) endorsed the use of PSMA-PET in patients with biochemical relapse following radical treatment; at the same time, there was a consensus against the use of PET in established metastatic PCa [8].

A Health Economic analysis of the added value of PSMA-PET to patient care identified a cost saving and, also, a more effective outcome in terms of life years saved [9]. Nevertheless, there is scope to improve the clinical application further and adopt AI to enhance the benefits.

## 3. Radiomics Applications

### 3.1. Introduction

Radiomics is based on the concept that standard-of-care medical images contain inherent information in the detailed structure of the image. It is believed that this inherent information is a result of the underlying tissue heterogeneity, including its micro-environment. Radiomics analysis involves the analysis of the relationships between voxels in the image, using the application of filters to extract specific feature classes which are then numerically analyzed to determine numerical characteristics, such as information content and repeatability of voxel intensities [10].

These features are often described as being invisible to the human eye [11] and hence, provide new information to the patient management workflow. The numerical description of the features makes them suitable for statistical and AI analysis. Commonly used radiomics features are texture measures and may be classified by the complexity of the relationships between voxel intensities [12].

First-order features analyze the histogram of intensities within a region, such as the tumor mass and include variance, skewness and kurtosis. Second- and higher-order features evaluate the relationships between pairs or groups of voxels. The most commonly used are the following: grey-level co-occurrence matrix (GLCM—the relationship between pairs of voxels); grey-level size zone matrix (GLSZM—the number of connected voxels with the same intensity), grey-level run length matrix (GLRLM—the number of connected voxels with the same intensity in a given direction); and neighboring grey tone difference matrix (NGTDM—the average difference between the intensity of a voxel and its neighbors). Other features include shape parameters (such as mean 2D and 3D radius) and fractal parameters (such as fractal dimension) [13]. Radiomics analysis is usually carried out using publicly available toolboxes such as Pyradiomics [14] and the package of Vallières [15] implemented in Matlab. There is much interest in bringing the image-based information from radiomics into the multiomics framework with the idea of combining biomolecular-level information with imaging. Zanfardino et al. [16] present a Multi Assay Experiment (MAE) approach to achieve this and present a case study based on MRI for breast cancer.

The clinical application of radiomics involves determining the relationship between the cohort of features extracted from the images and the clinical application of interest. In PCa, the common imaging modalities are MRI (including T1, T2 weighted and diffusion-weighted) transrectal ultrasound, conventional CT, conebeam CT and molecular imaging, often in the form of PET/CT, with tracers such as radiolabeled Prostate Specific Membrane Antigen (PSMA) [17] and fluorine labeled 18F-choline [18]. An analysis of the relationship between radiomics features has been carried out using standard statistical approaches as well as AI approaches.

### 3.2. Radiomics and AI

AI analysis is often carried out using supervised learning, in which the AI system is given the radiomics data as input, and the output is a prediction of the clinical application. The AI system is optimized until it achieves the desired sensitivity and specificity in predicting the answer to the clinical application. The AI methods used range from the simplest, such as the linear Support Vector Machine (SVM), to deep learning with multilayer neural networks. Outstanding challenges with radiomics include the following: (1) understanding the relationship between the signals generated from analysis of images with voxels of mm dimension and the underlying biological properties at the molecular level; (2) the scale of the problem, as studies usually have access to patient datasets in the hundreds and the number of radiomics features generated may be in the thousands, so there is an overfitting challenge; (3) related to 2, whether the radiomics dataset can be reduced a priori to reduce dimensionality of the problem. We now consider the clinical applications of radiomics in PCa.

In terms of imaging modalities, there are a number of reviews of radiomics in MR for PCa [5,19,20,21]. For this reason, this review will focus on radiomics in molecular imaging in PET and in CT.

### 3.3. Applications in PET

In molecular imaging in PET, studies have evaluated radiomics features of PSMA scans for lesion detection and characterization, assessment of Gleason score (GS), to characterize disease risk and for outcome prediction. Moazemi et al. [22] developed a radiomics model to differentiate pathological and physiological tracer uptake for Ga-labeled PSMA scans. The concept was to develop a tool to aid the radiologist in characterizing hotspots between intraprostatic lesions (IPLs) and normal prostatic tissue (NPT). Input data were 2419 hotspots from 72 patients. A total of 40 radiomics parameters were generated from the PET and low-dose CT scans. Erle et al. [23] further developed this work with data from 87 patients (72 for training and 15 with multiple hotspots for validation). With 77 radiomics features, they achieved a receiver operating characteristic (ROC) area under the curve (AUC) of 0.98 with 0.97 sensitivity and 0.82 specificity. Detection of IPLs was addressed by Zamboglou et al. [24]. The premise was that IPLs that may be missed by visual inspection might be detected by radiomics. Patient data consisted of a training set of 20 and an external validation set of 52 cases. Histology was used as the gold standard. A total of 154 radiomics features were used. In the training dataset, visual inspection missed lesions in 60% of the patients. Two radiomics features, based on analysis of local binary patterns (LBP), detected visually unknown lesions with an AUC of 0.93. For the validation set, visual inspection missed lesions in 50% of patients, but the LBP radiomics features yielded sensitivity values above 0.80. Domachevsky et al. [25] also discussed IPL versus NPT hotspot detection. They evaluated the suitability of PET PSMA SUVmax (maximum standardized uptake volume) and the apparent diffusion coefficient (ADC) from diffusion-weighted MRI as imaging biomarkers for distinguishing IPLs from normal tissue. Data from 22 patients yielded 22 IPLs and 22 NPTs. Results show significant statistical differences between IPLs and NPTs for SUVmax, ADCmin and ADCmean and conclude these are suitable for a radiomics model for lesion detection.

Alongi et al. [26] evaluated the use of radiomics with 18F-choline PET images for disease outcome prediction coupled with sub-group analysis by TNM staging. They analyzed data from 94 high-risk patients consisting of 18F-choline images for restaging and follow-up data. They extracted 51 radiomics features, using LIFEx software (v6.65) [27] and statistics-based feature reduction. For the whole dataset, two first-order histogram features were able to predict disease progression (67.6% accuracy). For sub-group analysis based on TNM staging, the numbers of features and accuracy were as follows: T:- 3 features, 87% accuracy; N:- 2 features, 82.6% accuracy; M:- 2 features, 72.5% accuracy. Risk stratification using the PSMA tracer DCFPyL was evaluated by Cysouw et al. [28]. In this prospective study, 76 medium- to high-risk patients who underwent radical prostatectomy and PSMA PET/CT had their primary tumor delineated and 480 radiomics features calculated per tumor. Random forests were trained with the radiomics features to model GS (≥8), lymph node involvement (LNI), metastasis and extracapsular extension (ECE), producing AUC values of 0.81, 0.86, 0.86 and 0.76, respectively. Using standard PET metrics produced lower AUC values of 0.76, 0.77, 0.81 and 0.67.

Yao et al. [29] evaluated the effect of outlining threshold as a percentage of SUVmax on prediction of GS, ECE and vascular invasion (VI) using LIFEx [26] and an SVM, for PSMA scans of 173 patients, divided into 122 training and 51 testing groups. Thresholds between 30% and 60% of SUVmax were evaluated. The optimum thresholds were as follows: for GS, 50% with AUC ≥ 0.80 for both training and test set; for ECE, 40% with AUC 0.77 and for VI, 50%with AUC 0.74. The recommendation was that SUVmax values of 40–50% are optimal for radiomics modelling of biological characterization of PCa. A second study by Zamboglou et al. [30] also addressed the use of radiomics to predict IPLs, GS and LNI from PSMA PET scans. They trained the model with 20 cases and validated it with a further 40 cases. Strong spatial correlations between histopathology and radiomics (>76%) showed the ability to distinguish between IPLs and NPT. A single texture feature distinguished between GS 7 and ≥ 8 (AUC = 0.91 prospective and 0.84 validation data). The same feature also distinguished between N1 and N0 nodal status (AUC = 0.87 prospective and 0.85 validation). Moazemi et al. [31] used pretherapeutic Ga-labeled PSMA scans to model overall survival of patients treated with 177Lu-PSMA. They retrospectively analyzed data for 83 patients with advanced PCa. The parameters used were 73 radiomics and 22 clinical features. The Cox proportional hazard and LASSO were used to select the most relevant features: SUVmin and kurtosis of the histogram. The Kaplan–Meier analysis was then used to evaluate the radiomics and clinical features for predicting outcome. Results showed that a radiomics signature based on SUVmin and kurtosis plus several other features showed a *p*-value < 0.05, supporting the hypothesis that radiomics using pre-therapeutic scans may be able to predict overall survival.

### 3.4. Applications in CT

Most uses of CT are not considered to be molecular imaging, but CT can provide tissue functional information, for instance, in contrast-enhanced CT. In addition, radiomics studies with CT often assume that the CT signal provides information on functions such as testing surrogacy for molecular imaging as in PET. Acar et al. [32] retrospectively modelled the responses of 75 patients treated for PCa with known bone metastasis. The clinical question was the use of CT radiomics for differentiation between metastatic lesions with PSMA expression and sclerotic regions that have responded to treatment (and no PSMA expression). A range of machine learning approaches were used including KNN, SVM and decision trees. Radiomics parameters were generated using LIFEx [27]. AUC values were between 0.63 and 0.76. The conclusion was that radiomics with machine learning can distinguish between metastatic lesions and sclerotic regions on CT scans. Peeken et al. [33] developed a model for lymph node metastasis detection using CT data. They used data for 80 patients who were treated with radio-guided surgery for resection of PSMA-positive metastases. 47 patients’ data were used for training and 33 for validation. Histological nodal status was used as a reference. A total of 156 radiomics features were extracted. The best radiomics model gave the best predictive performance (AUC = 0.95). This performed significantly better than conventionally used parameters such as lymph node short diameter. Bosetti et al. [34] studied risk stratification and biochemical relapse using weekly conebeam CT (CBCT) scans of 31 patients treated with radiotherapy. Pyradiomics [8] was used to extract radiomics features. 15 features were selected: histogram and shape-based. Logarithmic regression was used to predict tumor stage, GS, PSA, risk stratification and biochemical recurrence. Results showed AUC values of 0.78–0.80, 0.80–0.82, 0.83, 0.83 and 1.00, respectively.

Osman et al. [35] also evaluated CT scans for GS estimation and risk stratification. They used radiotherapy planning CT scans of 342 patients. The RadiomiX software (v3.0.1) used generated 1618 features, which were reduced to 522 after stability analysis. Their results distinguished between GS ≤6 and ≥7 with AUC = 0.90, and for GS 7, between 3 + 4 and 4 + 3 with AUC = 0.98. In terms of risk stratification, low- versus high-risk group distinction had an AUC = 0.96. Mostafaei et al. [36] studied the use of CT radiomics coupled with clinical and treatment (dose-volume) parameters to model toxicities in radiotherapy for 64 patients. The toxicities studied were urinary and gastro-intestinal (GI). Three sets of models were developed: radiomic (R—imaging only), clinical/dosimetric (CD) and radiomic/clinical/dosimetry (RCD). A total of 31 developed grade 1 or above GI and 52 urinary toxicities. For GI toxicity modelling, AUC values were 0.71, 0.66, 0.65 for R, CD and RCD, respectively, and for urinary, 0.71, 0.67 and 0.77, respectively. Tanadini-Lang et al. [37] demonstrated radiomics in CT perfusion scans for PCa with the specific aim of predicting tumor grade and aggressiveness. Data from 41 patients were analyzed. 1701 radiomics parameters were reduced to 10 using PCA (principal component analysis), which were then used in multi-variate analysis. Weak correlation was found between GS and radiomics parameters. The same parameter with the interquartile range of the mean transit time (MTT) from the perfusion scans was found to be useful for risk group prediction (AUC = 0.81). Two different radiomics parameters distinguished risk groups (AUC = 0.77).

### 3.5. Conclusions

To conclude this discussion on radiomics, radiomics shows much promise as a component of the AI toolbox in PCa imaging. There are many promising results from early studies, but challenges remain in this relatively new field, including reproducibility of results between centers and translation of models between patient groups. There is a need for larger independent datasets to test models. The reproducibility of radiomics features and their relationship to biomedicine needs further study [38], and it is important that methodologies are reported in sufficient detail to enable others to reproduce results [39].

## 4. Convolutional Neural Networks

### 4.1. Introduction

Convolutional neural networks (CNNs) were introduced in 1980 by Fukushima et al. [40]. The modern CNN consists of distinct layers, these being the convolutional layer, then the pooling layer. In the convolutional layer, a kernel, or filter, moves over the input data performing elementwise multiplication, effectively summing the results into a single output pixel. These convolutions create a feature map of the input, and these feature maps can highlight edges and irregularities within an image. The pooling layer takes the feature map generated by the convolutional layer and either takes the maximum value within the kernel or takes an average of the values within the kernel. This reduces the dimensionality of the image, allowing for less important information to be discarded but important features to be kept.

### 4.2. Applications

Classification of malignancy is a common application of AI to PCA. Many studies attempt to automatically determine if a patient will have malignancy or not, such as the work proposed by Hartenstein et al. [41], in which they develop a CNN to determine if 68Ga-PSMA-PET/CT-Lymph node status can be found via just the CT. The dataset consisted of 549 patients who received 68Ga-PMSA PET/CT scans. Three separate models were trained: for infiltration status, lymph node location and masked tumor locations. These models performed well with an area under the curve (AUC) of 0.95, 0.86 and 0.86, respectively. This was higher than the average uro-radiologist performance with an AUC of 0.81. Similar to this, Di Xu et al. [42] proposed a 2.5-dimension metastatic pelvic lymph node detection algorithm with CT only. This model achieved a sensitivity of 83.35% and an AUC of 0.90. Borrelli et al. [43] proposed a lymph node metastatic detection model for 18F-Choline PET/CT in 399 patients. This model outperformed a second human reader with 98 lesions detected, compared to the 87 lesions from the reader. Ntakolia et al. [44] proposed a lightweight CNN to classify bone metastasis. This model had fewer parameters than many other bone metastasis models, hence reducing the complexity of the model. The dataset consisted of 817 PCa patients with scintigraphy scans. This model managed to achieve a sensitivity of 97.8% and specificity of 98.4%. Full classification accuracy was 97.41%.

Capobianco et al. [45] proposed a dual tracer learning model on 173 patients with 68Ga-PSMA-11 PET/CT to classify full-body uptake in PCa patients. This was carried out by passing the patient data through a CNN to classify sites of elevated tracer uptake as either suspicious or non-suspicious. These results were then assigned to an anatomical region. In total, of the 5577 high-uptake regions which were annotated, 1057 were suspicious. The model provided an average accuracy of 78.4% for suspicious regions, and a 94.1% accuracy for all uptake. Tragardh et al. [46] proposed a primary tumor and metastatic disease analysis model for whole-body 18F-PSMA1007 PET/CT. A total of 660 patients were analyzed with full-body PET/CT scans. This model managed to provide a sensitivity of 79% for detecting lesions, 79% for lymph node metastasis and 62% for bone metastasis. This was in contrast to the nuclear medicine physician sensitivities of 78%, 78% and 59%, respectively.

Jong Jin Lee et al. [47] proposed a recurrence detection algorithm with 18F-fluciclovine PET. Three models were trained. One model was trained with a single slice approach, one trained with a 2D case-based approach and one with a full 3D approach. The dataset consisted of 251 patients with PET scans labeled as either normal, abnormal or indeterminable. The 2D CNN slice-based approach had a sensitivity of 90.7% and a specificity of 95.1% with an AUC of 0.97. The 2D case-based approach achieved a sensitivity of 85.7% and a specificity of 71.4% with an AUC of 0.75. The 3D case-based approach achieved a sensitivity of 71.4%, a specificity of 71.4% and an AUC of 0.70.

These models show that many models perform comparably to or sometimes better than professional radiologists showing the potential predictive power that CNNs can have for detecting and classifying a variety of problems in PCa, from tumor classification to metastatic location classification. This analysis highlights that while these models perform well, there is potential for larger datasets from different centers to be able to verify these models.

Compared to classification, segmentation classifies individual pixels based on a given mask; pixels inside the mask are classified as positive, while outside, they are classified as negative. With many PCa problems, there are much more negative pixels than positive which makes the problem difficult. Kostyszyn et al. [48] proposed a CNN to segment the Gross Tumor Volume (GTV) in primary PCa patients. The dataset had 209 patients from three separate centers who had PSMA-PET. The model utilized the 3D U-Net architecture to segment the GTV volume. The median Dice similarity scores for different center cohorts were 0.84, 0.81 and 0.83. Sensitivity and specificity were 98% and 76% for the first cohort and 100% and 57% for the second cohort. Wang et al. [49] proposed a dual attention mask R-CNN on PET/CT to segment prostate and the dominant intraprostatic lesions. The dataset consisted of 25 patients with PET/CT scans. The first network of the pair attempted to locate a rough, initial ROI in the patient, while the second performed the segmentation. The model had a Dice similarity score of 0.84 ± 0.09 (SD—1 standard deviation).

Matkovic et al. [50] similarly proposed a cascaded regional-net for prostate and dominant intraprostatic lesions. This model used 49 patients and achieved a Dice similarity score of 0.93 ± 0.06 for the prostate and 0.80 ± 0.18 for dominant intraprostatic lesions. Rainio et al. [51] proposed a method of choosing separate thresholds for each PET image slice with a CNN to label pixels directly on the slice. This was combined with a CNN which used constant thresholding to pick the optimal thresholds. The dataset consisted of 78 PCa patients. The average Dice similarity score was 0.72 ± 0.41 for the variable thresholding, and 0.69 ± 0.38 for the mixed method. Holzschuh et al. [52] proposed a CNN trained on 18F-PSMA-1007 PET to segment intraprostatic GTVs in primary PCa. The model was trained on 128 patients, tested on an independent internal cohort of 52 patients and externally validated on three datasets with 14, 9 and 10 patients with different radiotracers. The median dice scores were 0.82 for the internal set, 0.71 for an external set with the same tracer, 0.80 for the external set with 18F-DCFPyL-PSMA and 0.80 for the external set with 68Ga-PSMA-11.

Zhao et al. [53] proposed a classification and segmentation model on 68Ga-PSMA-11 PET/CT images. There were 193 patients in the cohort trained on triple-combining a 2.5D U-Net. The network performed well with a precision, recall and F1 score of 0.99 for bone lesion detection and 0.94, 0.89 and 0.92 for lymph node detection. Segmentation achieved average dice scores of 0.65 and 0.55 for bone and lymph node lesions, respectively. Ghezzo et al. [54] externally validated a CNN trained to segment prostate GTVs. This model was validated on 68Ga-PSMA-PET images, with 85 patients being included in the dataset. The model achieved a median dice score of 0.74, and the models were robust between modalities and ground truth labels. These segmentation models show that it is entirely feasible to use CNNs to segment sites of risk in PCa. As mentioned in the classification section, there is a need for larger scale datasets and models to be trained, and the black box nature of CNNs does not lend well to explaining how the models come to their conclusions.

## 5. Unsupervised Learning

Most models in clinical applications tend to be supervised, as they are performing some diagnostic or segmentation task and are trained using ground truth information. Unsupervised methods potentially have greater clinical utility as they can use the large volumes of available data, without the need for time consuming and costly-to-acquire expert labeling. Some unsupervised models have been developed or utilized for traditionally supervised tasks; however, they still require labeled data for validation of model outputs as is the case for supervised models.

Unsupervised learning is a crucial tool in AI for pattern recognition, and this can provide powerful tools for knowledge discovery in medical data. In the medical context, these methods would primarily be used to cluster or subtype patients, learn latent features in the inputs that may be used for downstream supervised tasks [55,56,57,58,59], such as feature selection, computer-aided detection [60] or for visualization of data and machine learning (ML) results [61]. One core methodology in unsupervised learning is the use of clustering methods to obtain group membership of data based on selected input features [55]. The use of clustering in medical imaging is extensive, with applications in areas such as segmentation in CT colonography [62] and 3D segmentation [59]. Dimensionality reduction methods have also been used in AI workflows, such as in the visualization of labeled data and to model outputs such as risk scores [63].

An emerging trend is the use of generative AI in medical imaging for tasks such as modality transfer or to increase the number and variation of images in the dataset [64]. Here, typically, you have pairs of images where the matching between images constitutes a label. Modality transfer is a growing area of interest, but the clinical utility is limited due to barriers around validation and verification of the synthesized images. Instead, most uses of generative models are for increasing the available data for training models or in the training of adversarial models such as GANs [64,65].

The emergence of transfer learning has enabled applications with small amounts of data to access deep learning models trained on very large datasets from a different domain. Typically, this is used in supervised tasks, but it also enables the use of the pre-trained network as a feature extractor for clustering or classification tasks. The performance of transfer learning models demonstrates their potential impact in clinical tasks [66,67,68], such as lesion detection [69] and combining patient imaging and demographics data [70]. Pretrained models transferred from one task to another have been shown to perform at least as well as models trained (from random network weights) for the specific task, and fine-tuning these models improves robustness [69].

The so-called next generation of AI is the use of foundational models, where a model is trained on many diverse datasets in order to perform multiple tasks [71]. These can help generalize models and reduce the need to have large amounts of labeled data in a specific domain [72,73], and these have been used for analysis of patient records [74,75] and classification of pathologies [76], with potential for patient support systems [77,78].

## 6. Semi-Supervised Learning

### 6.1. Introduction

In the domain of deep learning, limited labeled datasets can lead to overfitting, emphasizing the need for high-quality patient data to minimize bias in clinical practice. Nevertheless, addressing this challenge is complicated by privacy and ethical concerns surrounding medical data. Moreover, the scarcity of labeled data for training deep learning algorithms makes manual labeling both expensive and reliant on physician expertise. A potential mitigating solution to this issue is semi-supervised learning (SSL). SSL improves model performance by integrating a restricted set of labeled data with a more extensive pool of unlabeled data.

Present SSL solutions are structured around three fundamental assumptions [79]: (1) smoothness, positing that similar images share similar labels; (2) low-density, asserting that decision boundaries avoid high-density regions in the feature space; and (3) manifold, contending that samples on the same low-dimensional manifold within the feature space bear the same label. Exploring these assumptions, SSL methods can broadly be categorized into pseudo-label-based SSL [80,81,82] and consistency-based SSL [83,84,85].

In pseudo-labeling SSL [80,81,82], the model’s predictions on unlabeled data serve as additional virtual labels, providing a form of supervision for the model to learn from the unlabeled samples. This helps in leveraging the information present in the unlabeled data to improve the model’s performance. However, pseudo-labeling SSL assumes that the model’s predictions on unlabeled data are reasonably accurate, which means that if the model’s predictions on unlabeled data are noisy or unreliable, it may negatively impact the overall performance. Therefore, careful consideration and validation of the pseudo-labeling process are crucial for successful implementation. On the other hand, consistency learning [83,84,85] exposes the model to perturbed versions of the same input and encourages it to provide consistent predictions, so it can learn more robust and generalizable representations. This approach leverages the unlabeled data to encourage the model to capture the underlying structure of the data, which can lead to improved performance, especially when labeled data are scarce. Consistency learning typically exhibits higher accuracy than pseudo-labeling. This discrepancy can be attributed to the fact that pseudo-label methods overlook a portion of the unlabeled training dataset during the training process, consequently diminishing their generalization capabilities.

Given the superior performance of consistency learning, we now focus on the main approaches proposed in the field. The Π model [83] introduces a weak-strong augmentation scheme, generating pseudo labels through weakly augmented data while making predictions based on its strongly augmented version, incorporating noise perturbation or color-jittering. To enhance the stability of these pseudo labels, the Π model variant, Temporal Embedding [83], employs an exponential moving average (EMA) method, accumulating historical results. Despite its stability improvement, Temporal Embedding incurs high hardware costs associated with storing historical results.

Addressing this issue, Mean Teacher (MT) [85], a widely adopted semi-supervised structure, overcomes the challenge by ensembling network parameters with a “teacher” network through EMA transfer to produce pseudo labels. However, a notable drawback of consistency-learning methods, including Mean Teacher, is their tendency to converge to the same local minimum during training [81], resulting in both teacher and student models exhibiting similar behavior for many complex data patterns.

### 6.2. Applications

In medical image segmentation, there has been a notable surge in interest in semi-supervised learning, particularly with a focus on segmenting uncertain regions within pseudo labels. Recent work [86,87,88] introduces uncertainty-aware mechanisms that utilize Monte Carlo dropout to estimate uncertainty regions associated with pseudo labels. Others, such as [89,90], progressively learn unsupervised samples by considering prediction confidence or the dissimilarity between predictions from different scales. An alternative approach by [91] involves estimating uncertainty through a threshold-based entropy map, while [92,93,94] gauge uncertainty by calibrating predictions from multiple networks.

While these methodologies aim to avoid potential noise from unlabeled data, they unintentionally sideline the learning of potentially accurate pseudo labels, leading to insufficient convergence, especially in the face of complex input data patterns. Addressing this concern, ref. [95] proposes a complementary learning approach, leveraging negative entropy based on inverse label information. Refs. [96,97,98] successfully explore both negative and positive learning techniques to strike a balance between steering clear of learning from potentially noisy regions and mitigating insufficient convergence.

In pursuit of good generalization, certain studies [99,100,101] employ adversarial learning, while others [102,103,104,105] leverage perturbations across multiple networks to enhance consistency. Despite their success in recognizing unlabeled patterns, these works have overlooked the importance of in-context perturbations in input data, crucial for prompting the model to discern spatial patterns. Addressing this gap, ref. [106] has introduced a contrastive learning framework aimed at capturing such spatial information in the segmentation of urban driving scenes. However, their contrastive learning framework relies on potentially erroneous segmentation predictions, introducing a risk of confirmation bias in the pixel-wise positive/negative sampling strategy, leading to suboptimal accuracy in handling complex medical images. Additionally, their framework does not explore network perturbations, thereby limiting its generalization capacity.

In the specific area of single-photon emission computerized tomography (SPECT), Apiparakoon et al. [107] applied a semi-supervised learning (SSL) technique known as the Ladder Feature Pyramid Network (LFPN). LFPN is distinctive for incorporating an autoencoder structure within the ladder network, enabling self-training using unlabeled data. While LFPN, when utilized in isolation, achieves a slightly lower F1-score compared to self-training methods, it significantly outperforms in terms of efficiency, demanding only half the training time.

Moreover, in addressing the challenge of limited labeled data, alternative strategies have been proposed. Some studies advocate for pretraining the model by leveraging unlabeled data obtained from related datasets, as suggested in prior research [108,109]. This pretraining approach seeks to enhance the model’s performance by leveraging additional information from related datasets, thereby compensating for the scarcity of labeled data in the specific domain of interest, such as SPECT imaging.

## 7. Generative Neural Networks

### 7.1. Introduction

Generative adversarial networks (GANs) are a type of machine learning architecture which consists of two models which play a game [110], Figure 1. One network is the generator defined as pmodel(x), which can draw samples from the distribution pmodel. Generators are defined by a prior distribution *p*(z) over a vector z which is the input to the generator function: G(z; θ(G)), where θ(G) is a set of learnable parameters which define the generators strategy for the game. The input vector z is a source of randomness, similar to a seed in a pseudorandom number generator. The prior distribution *p*(z) is usually an unstructured distribution. Therefore, the goal of the generator is to learn the function G(z) that transforms the unstructured noise z into realistic samples. The other network is the discriminator. The discriminator identified samples x and returns an estimation Dx; θ(D) of whether x is taken from the real dataset, or taken from the pmodel generated by the generator. Each model has a cost function: JG(θ(G), θ(D)) for the generator and JD(θ(G),θ(D)) for the discriminator. The discriminator’s cost encourages it to classify the real data correctly, and the generator’s cost encourages it to generate samples that the discriminator will incorrectly classify.

### 7.2. Applications

The clinical applications of generative models are wide, as shown in Table 1, ranging from Dosimetry prediction, tumor segmentation and dose plan translation.

Xue et al. [111] proposed a 3D GAN model to predict the required post-therapy dosimetry on patients with metastatic castration-resistant PCa (mCRPC). The dataset consisted of 30 mCRPC patients, with 48 treatment cycles with 68Ga-PSMA-11 PET/CT. The model utilized a 3D U-Net as the generator and a standard CNN as the discriminator; a further dual-input model was created which incorporated the CT and PET information. The model also used voxel-wise loss alongside an image-wise loss function for better synthesis. Results from this model consist of a voxel-wise mean absolute percentage error of 17.57% ± 5.42% (one standard deviation), and the dual input model achieved a mean absolute percentage error of 18.94% ± 5.65%.

Murakami et al. [112] proposed an architecture known as a Pix2Pix GAN to translate CT and structure sets into a fully automated dose plan. The motivation for this was to reduce the need to annotate patient structures, which is required for previous dose prediction models. The dataset consisted of 90 IMRT (intensity modulated radiotherapy) plans of PCa patients’ treatments. Each of these patients was prescribed 78 Gy in 39 fractions to a planning target volume (PTV) which could be covered by 95% of the prescribed dose. The PTV was generated by adding 5 mm margins to the clinical target volume (CTV). The matrix size of the RT-Dose array was converted to 512 × 512 images with 16 bits to match the CT image size. The Pix2Pix model focuses on generating synthetic images from a source to a target image, in this case, CT and structures sets to RT-dose images. The final dose differences were within approximately 2% of the PTV (except D98% and D95%) and approximately 3% for organs at risk for the CT-based dose prediction. The structure-based approach was within 1% of the PTV and 2% for the organs at risk. Mean prediction time for the model was approximately 5 s.

Sultana et al. [113] proposed a 3D U-net Generator and fully connected network discriminator architecture to segment the region of interest in PCa patients. The dataset consisted of 115 PCa patients with CT scans of the pelvic region. Manual contours of the prostate, bladder and rectum were drawn by a radiation oncologist. The CT scans were down-sampled from dimensions 512 × 512× [80–120 slices] to 118 × 188 × [48–72 slices] on which the coarse segmentation model was trained. Following this, a fine segmentation model was trained on the ROIs generated by the coarse segmentation model. This model had state-of-the-art performance with a Dice similarity coefficient (DSC) of 0.90 ± 0.05 for the prostate, 0.96 ± 0.06 for the bladder and 0.91 ± 0.09 for the rectum. Zhang et al. [114] proposed an ARPM-Net for the prostate and the organ-at-risk segmentation in pelvic CT images. The dataset consisted of 120 patients with CT scans and structure sets. The segmentation network followed a U-net structure with added Markov Random field blocks which reduced the parameters of the model by half plus local convolutions. Results for this achieved higher DSC scores than the baselines generated in the model, giving the following: prostate with a DSC of 0.88 ± 0.09, bladder at 0.97 ± 0.08, rectum at 0.86 ± 0.08 and the left and right femurs both at 0.97 ± 0.01.

Heilemann et al. [115] proposed a U-Net CycleGAN to segment various tumors, including prostate. The prostate dataset consisted of 308 patients; however, models were created of various patient cohorts with the best models being tested on 179 patients. Structure sets and organs at risk were delineated by an experienced radiation oncologist. The U-net and GAN combination provided good DSC scores: rectum had a DSC of 0.84 ± 0.08, bladder 0.89 ± 0.08 and the left and right femurs both at 0.93 ± 0.06.

Chan et al. [116] proposed a variety of Cycle-GAN architectures to remove under-sampling artifacts and correct the image intensities of 25% dose CBCT images, creating a synthetic planning CT. The dataset consisted of 41 PCa patients who received volumetric modulated arc therapy (VMAT). Unpaired 4-fold validation was used to enable the median of four models to be used for the model output. The DSC for the anatomical fidelity of the bladder achieved an average test score of 0.88 with the standard Cycle-GAN and an average score of 0.92 for a residual GAN. The rectum achieved an average score of 0.77 for the standard GAN and a score of 0.87 for the residual GAN.

Pan et al. [117] proposed a diffusion probabilistic model to automatically synthesize a variety of different medical images. Diffusion models are an alternative approach to image generation; these start by gradually adding noise to the dataset and train a model to learn to denoise the images. As the model denoises the images from random noise, the model effectively learns how to generate samples that are purely synthetic and are not a part of the original cohort. The cohort for the prostate segment of their study consisted of scans of 93 patients, totaling 4157 2D CT slices in total. These were passed through the diffusion model and then subsequently classified into pelvic or abdomen CT with an accuracy of 89% and sensitivity of 82% for the fully synthesized dataset, and 93% accuracy and 90% sensitivity for a mix of synthetic and real data.

## 8. Discussion

The role of bio-imaging, functional and anatomical imaging is essential to the successful management of PCa. In terms of imaging with molecular-based signatures, MR plays a pivotal role. There are a range of review articles that discuss the role of MR and the application of ML and AI to MR [6,21,118,119,120], and the reader is referred to these for more information on this area. PET/CT molecular imaging is also pivotal and is playing a growing role, particularly with the advent of PSMA and the availability of new radioisotopes and radiopharmaceuticals. For these reasons, this review has focused on PET/CT imaging of PSMA.

Outstanding challenges in AI in biomolecular imaging for PCa include the role of the clinician. It is the view of the authors that AI will be used as a tool to support the clinical decision process, particularly providing support for borderline cases. Furthermore, the role of AI decision systems will evolve particularly as the AI systems become more accurate. Dataset size in cancer is small in AI terms. Often, the original algorithms are demonstrated on datasets of hundreds of thousands of images, whereas the imaging datasets contain smaller numbers, often one to several hundreds. This can be ameliorated by the choice of training strategy as discussed above. A second unique feature of cancer imaging datasets, particularly in screening and disease detection, is that the data may be imbalanced, for example, with a predominance of disease-free scans. The approaches to ameliorate this problem are also discussed above. Other challenges include bias in the dataset. If the AI model is trained on data from one population or gender, then it may be less suitable for another population. Explainability is a challenge in many AI applications but is especially important in healthcare. If an AI system predicts poor outcome, for instance, what are the features in the data that prompt this prediction? This is not only important in quality assuring the results of the system but may have an impact in changing a poor prognosis to a good one.

The shape of AI in the ‘real-world’ clinical practice is a contentious point. A total of 276 radiologists from different countries were asked regarding their experience in integrating AI tools, and only 17.8% experienced difficulties. It was seen in 23.4% of the cases that any benefit reduces the workload, but only 13.3% of radiologists would like to invest in AI tools. Hence, we are seeing some skepticism amongst stakeholders [121]. Other challenges to the use of AI in this area include the requirement for a regulatory framework that encompasses the use of AI, how the role of AI is incorporated into the patient management workflow, considerations of commercialization, ownership of data and how the development of the AI system is managed. For example, Zhang et al. [122] identified five subject areas which influence the trustworthiness of medical AI. Considerations of these five subject areas will allow for easier integration of AI into the clinic.

A range of groups are making available cancer-imaging datasets to aid development of machine learning and AI systems in cancer [123]. This is in its infancy, and these datasets will provide a gold standard with which to compare different algorithms to test their relative strengths and weaknesses and to support the future development of new AI methods. Furthermore, given the need for large datasets for AI systems and the need for sufficient heterogeneity in the dataset to avoid bias and misdiagnosis of smaller class groups, there is a strong need for multi-center collaboration to develop large datasets and to enable data-sharing. This will also enable the development of future AI systems.

## 9. Conclusions

This review has discussed the role of ML and AI in molecular imaging for PCa with an emphasis on PSMA-based PET/CT. This is a rapidly evolving field with many clinical application papers cited from the past few years and rapidly developing AI approaches.

The current status is that there are a range of studies demonstrating the use of AI in PET/CT for PCa which demonstrate a range of clinical measures such as lesion detection, outlining and outcome prediction. In many cases, these produce as-good-as or better results than the human, but the studies are often small in scale and on a single patient population. There is the need to upscale such studies to larger, more heterogeneous populations. The growing availability of new AI methodologies that are tailored to smaller datasets and unlabeled and semi-labeled data show great promise for impact in this area. In conclusion, current studies provide strong evidence for the role of AI in this area. The next challenge is to make these mainstream, addressing challenges such as integration into the clinic, dataset size and optimal AI algorithm.

## Figures and Tables

**Figure 1 jpm-14-00287-f001:**
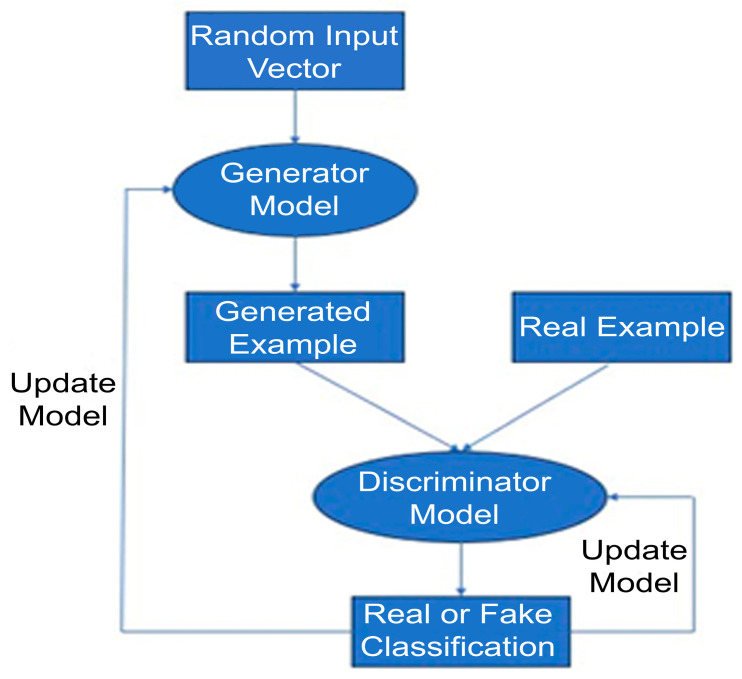
Typical training loop for GANs.

**Table 1 jpm-14-00287-t001:** GAN-related studies on CT and PET/CT.

Study	Cohort	Model	Application
Xue et al. [111]	30 Patients–48 PET/CT treatment cycles	3D-RLT-Dose-GAN	Voxel-wise Prediction of Post-therapy Dosimetry
Murakami et al. [112]	90 Patients	Pix2Pix GAN	Translates CT and structures to a dose plan prediction
Sultana et al. [113]	115 Patients	3D U-Net and FCN GAN	Segmentation of the ROI via GAN
Zhang et al. [114]	120 Patients	ARPM-Net	Multi-Organ Segmentation on the pelvic area
Heilemann et al. [115]	Multiple Datasets	U-net and U-Net CycleGAN	Segmentation
Chan et al. [116]	41 Patients	Modified CycleGAN	low dose calculation
Pan et al. [117]	94 CT patients	Diffusion Probabilistic Model	Medical image synthesis

## Data Availability

No new data were created or analyzed in this study. Data sharing is not applicable to this article.

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
