# Peer review of "The Application of Radiomics and AI to Molecular Imaging for Prostate Cancer"

_jpm, 2024, doi:10.3390/jpm14030287_

Round 1
Reviewer 1 Report
Comments and Suggestions for Authors
The authors should be congratulated for trying to provide an interesting manuscript. However, some crucial points warrant mention:
- the English form should be revised. Some sentences are written difficulty, typos are also present (i.e. PSAM line 76);
- the aim of the study is not clear in the "Introduction" section, why the scientific literature need this review?
Moreover, the methods are not clear either, is it a systematic or a narrative review?
- line 79, EAU is not the European Association of Radiology;
- solve the difference in lines 88 and 94;
- I suggest to re-organize the manuscript into subsections. The topic is complex and it is really difficult to follow all the information put together;
similarly, the authors are invited to shorten paragraphs and organize better all the information provided.
- line 131 what does "related to ii," mean?
-the discussion section is not understandable, not providing any improvement in the understanding of reported studies. Similarly, the conclusion paragraph is hastened and should be re-elaborated.
Comments on the Quality of English LanguageEnglish form should be completely revised.
Author Response
Dear Editor and Reviewers,
I am pleased to resubmit for publication the revised version of jpm-2855976 manuscript, entitled “The Application of Radiomics and AI to molecular Imaging for Prostate Cancer”.
Thankfully the reviewers provided us with a great deal of guidance, regarding how to better position the article. We are hopeful you agree that this revision will update our comprehensive review. All the comments have been addressed, as shown in the revised version of the manuscript, along with this point-by-point response to the reviewers' comments.
All corresponding are blue changes in the manuscript.
Reviewer #1:
-
General comment:
“The authors should be congratulated for trying to provide an interesting manuscript.”
Response:
Thank you for your positive reinforcement and constructive feedback. We appreciate the opportunity to revise our work for consideration for publication.
-
Specific comments:
“However, some crucial points warrant mention:”
- the English form should be revised. Some sentences are written difficulty, typos are also present (i.e. PSAM line 76);
Response:
Thank you for your comment.
Typos have been corrected, for instance, we have corrected inconsistencies in the naming of PET/CT and 18F-choline. Moreover, PSMA in line 84 of the revised manuscript has now been corrected.
Our understanding from the comment on English form is that you recommend subsections and breaking up of the longer paragraphs. This has been done. The text has been broken down into smaller paragraphs particularly in sections 3 and 4, “Radiomics Applications” and “Convolutional Neural Networks”. Subsections have been added to the main sections 3,4,6 and 7.
- the aim of the study is not clear in the "Introduction" section, why the scientific literature need this review?
Response:
We have emphasized the need for this review in lines 57-60: “MRI has been extensively covered in previous review articles, there is a lack of a detailed review of PET-CT within PCa, in addition the role of both PET-CT and AI in prostate cancer imaging is rapidly growing. For these reasons this article will review the use of PET-CT and AI.”
Moreover, the methods are not clear either, is it a systematic or a narrative review?
Response:
This is a narrative review. We have clarified this in line 61: “This narrative review will provide an overview of imaging for PCa, with a focus on PET-CT and PSMA-PET, with comments on other modalities such as, MRI and ultrasound and on histopathology where appropriate”
- line 79, EAU is not the European Association of Radiology;
Response:
We have edited the abbreviation EAU to European Association of Urology (line 87 of the revised manuscript).
- solve the difference in lines 88 and 94;
Response:
The sentence starting line 96 of the revised manuscript has been edited to remove the double reference to features not visible to the human eye, as follows: “Radiomics is based on the concept that standard-of-care medical images contain inherent information in the detailed structure of the image.”
- I suggest to re-organize the manuscript into subsections. The topic is complex and it is really difficult to follow all the information put together;
Response:
Thank you for your comment.
Subsections have been added to the main sections 3,4,6 and 7.
similarly, the authors are invited to shorten paragraphs and organize better all the information provided.
Response:
Thank you for your comment.
The text has been broken down into smaller paragraphs particularly in sections 3 and 4. The subject matter is organized with a section on each methodology with an introduction to the method and then its application. This has been made clearer with the use of titled subsections.
- line 131 what does "related to ii," mean?
Response:
ii has been changed to 2 as it relates to item 2 on the list (line 144 of the revised manuscript).
-the discussion section is not understandable, not providing any improvement in the understanding of reported studies. Similarly, the conclusion paragraph is hastened and should be re-elaborated.
Response:
Thank you for your comment.
Two technical paragraphs have been removed from the discussion and other parts rewritten. A separate conclusion has been added.
English form should be completely revised.
Response:
Apologies; we have revised the English of the manuscript as you kindly requested. Extensive revision has been made to the manuscript including paragraph and sub-section restructuring and corrections.
Reviewer 2 Report
Comments and Suggestions for Authors
The work provides a good overview of AI techniques for prostate cancer molecular imaging, especially PET/CT. The manuscript is well organized, moving from an introduction to PSMA imaging and systematically reviewing major AI methods and application areas. A comprehensive set of relevant references are cited. Both methodology papers and application studies are covered. The pros and cons of different techniques are discussed. Challenges and future directions are summarized well.
Overall, this well-written review comprehensively summarizes the state-of-the-art in this emerging application area for AI in medical imaging. The work offers a significant contribution as a reference for prostate cancer imaging and AI researchers.
I have a few minor comments:
- The introduction could provide more background on the clinical need and potential impact of improving prostate cancer imaging with AI. This would strengthen motivation.
- Expanding on the barriers to clinical translation beyond just clinician skepticism may be beneficial. Issues like regulatory approval, integration with workflows, and commercialization could be mentioned
- Comparing the performance of different techniques on standardized datasets could better highlight limitations and optimization opportunities.
- Highlighting opportunities for multi-site collaboration and data sharing for larger datasets could encourage further progress in this area.
- A section on overall conclusions and future outlook could concisely summarize the current state and most promising directions in the field.
Author Response
Dear Editor and Reviewers,
I am pleased to resubmit for publication the revised version of jpm-2855976 manuscript, entitled “The Application of Radiomics and AI to molecular Imaging for Prostate Cancer”.
Thankfully the reviewers provided us with a great deal of guidance, regarding how to better position the article. We are hopeful you agree that this revision will update our comprehensive review. All the comments have been addressed, as shown in the revised version of the manuscript, along with this point-by-point response to the reviewers' comments.
All corresponding are blue changes in the manuscript.
Reviewer #2:
-
General comment:
“The work provides a good overview of AI techniques for prostate cancer molecular imaging, especially PET/CT. The manuscript is well organized, moving from an introduction to PSMA imaging and systematically reviewing major AI methods and application areas. A comprehensive set of relevant references are cited. Both methodology papers and application studies are covered. The pros and cons of different techniques are discussed. Challenges and future directions are summarized well.
Overall, this well-written review comprehensively summarizes the state-of-the-art in this emerging application area for AI in medical imaging. The work offers a significant contribution as a reference for prostate cancer imaging and AI researchers.”
Response:
Thank you for your positive reinforcement and constructive feedback. We appreciate the opportunity to revise our work for consideration for publication.
-
Specific comments:
“I have a few minor comments:”
-
The introduction could provide more background on the clinical need and potential impact of improving prostate cancer imaging with AI. This would strengthen motivation.
Response:
Thank you for your comment.
We have added the paragraph: “The scope for the used of AI in the PCa imaging pathway includes helping early detection and staging of disease, plus the prediction of outcome based on complex patterns in previous data. AI has the potential to support the oncologist in decision-making by providing extra analytical insight also flagging features in unusual cases. In summary AI has the potential to provide decision support tools and to support labor-saving.”, (lines 47-52 of the revised manuscript).
-
Expanding on the barriers to clinical translation beyond just clinician skepticism may be beneficial. Issues like regulatory approval, integration with workflows, and commercialization could be mentioned
Response:
Thank you for your comment.
We agree and have added the following: “Other challenges to the use of AI in this area include the requirement for a regulatory framework that encompasses the use of AI, how the role of AI is incorporated into the patient management workflow, considerations of commercialization, ownership of data and how the development of the AI system is managed.”, (lines 624-628 of the revised manuscript).
-
Comparing the performance of different techniques on standardized datasets could better highlight limitations and optimization opportunities.
Response:
Thank you for your comment.
Agreed. This point has been addressed, as follows: “A range of groups are making available cancer imaging datasets to aid development of machine learning and AI systems in cancer. This is in its infancy and these datasets will provide a gold-standard with which to compare different algorithms to test their relative strengths and weaknesses and to support the future development of new AI methods.”, (lines 629-633 of the revised manuscript).
-
Highlighting opportunities for multi-site collaboration and data sharing for larger datasets could encourage further progress in this area.
Response:
Thank you for your comment.
Agreed. We have added the following: “Furthermore given the need for large datasets for AI systems and the need for sufficient heterogeneity in the dataset to avoid bias and misdiagnosis of smaller class groups, there is a strong need for multi-center collaboration to develop large datasets and to enable data-sharing. This will also enable the development of future AI systems.”, (lines 633-637 of the revised manuscript).
-
A section on overall conclusions and future outlook could concisely summarize the current state and most promising directions in the field
Response:
Thank you for your comment.
An overall conclusions section has been added, (lines 638-651 of the revised manuscript).
Reviewer 3 Report
Comments and Suggestions for Authors
This review by William Tapper and co-authors provides an overview of AI applied to PET/CT in PCa. Overall, the review is well-executed.
- While the diagnosis of prostate cancer is not the main focus of the manuscript, the authors should introduce generic concepts to enable a wider audience of readers to understand the article.
- The authors should delve more deeply into the role of AI in PET/CT for PCa in guiding treatment decisions and treatment methods. They should also highlight the diagnostic advantages of AI in PET/CT for PCa compared to conventional PET/CT.
While the overall language quality of this manuscript is good, the authors should still carefully review the entire text before publication.
Author Response
Dear Editor and Reviewers,
I am pleased to resubmit for publication the revised version of jpm-2855976 manuscript, entitled “The Application of Radiomics and AI to molecular Imaging for Prostate Cancer”.
Thankfully the reviewers provided us with a great deal of guidance, regarding how to better position the article. We are hopeful you agree that this revision will update our comprehensive review. All the comments have been addressed, as shown in the revised version of the manuscript, along with this point-by-point response to the reviewers' comments.
All corresponding are blue changes in the manuscript.
Reviewer #3:
-
General comment:
“This review by William Tapper and co-authors provides an overview of AI applied to PET/CT in PCa. Overall, the review is well-executed.”
Response:
Thank you for your positive reinforcement and constructive feedback. We appreciate the opportunity to revise our work for consideration for publication.
-
Specific comments:
“While the diagnosis of prostate cancer is not the main focus of the manuscript, the authors should introduce generic concepts to enable a wider audience of readers to understand the article.”.
Response:
Thank you for your comment.
The focus of the paper is on AI and its application in prostate cancer imaging. We tried to introduce simple concepts, but we are also targeting a specialist audience.
“The authors should delve more deeply into the role of AI in PET/CT for PCa in guiding treatment decisions and treatment methods. They should also highlight the diagnostic advantages of AI in PET/CT for PCa compared to conventional PET/CT.”.
Response:
Thank you for your comment.
This review is focused on PET and CT. The treatment choice question is not solely an AI-guided one but will bring in other modalities and hence is not within the main scope of this article.
“While the overall language quality of this manuscript is good, the authors should still carefully review the entire text before publication.”
Response:
Extensive revision has been made to the manuscript including paragraph and sub-section restructuring and corrections. For instance we have corrected inconsistencies in the naming of PET/CT and 18F-choline. Typos such as the naming of PSMA in line 84 of the revised manuscript have now been corrected. Also in formatting some words which wrap around the end of lines have been hyphenated incorrectly by the automatic type-setting. These have been corrected.
Round 2
Reviewer 1 Report
Comments and Suggestions for Authors
Thanks for improving the manuscript according to my suggestions.
however, some other points warrant mention:
- all the added information misses of related citations.
- In lines 47-51, do the authors think that AI could be helpful "only" to oncologists in predicting PCa and outcomes?
- imaging does not diagnose PCa. The diagnosis is made by urologists by biopsies.
- I don't think that sentences like ". See for instance Delgadillo et al. [16] in relation to radiotherapy, Ferro et al. [17], 147 Penzkofer et al. [18] and Sun et al. [19]. For this reason, this review will focus on radiomics 148 in molecular imaging in PET and in CT." should be included in a scientific manuscript.
- with "Ga" do the authors refer to Gallium?
- doesn't seem polite to use "Introduction" and "Conclusions" as titles of subsections.
- the entire manuscript is not well-organized. As mentioned in the previous review step, some sections are too long and some sentences are useless. thus the manuscript does not provide a high-quality overall overview of the topic, useful to the readers.
Comments on the Quality of English LanguageImprovements are still needed.
Author Response
Dear Editor and Reviewer 1,
I am pleased to resubmit for publication the second round revised version of jpm-2855976 manuscript, entitled “The Application of Radiomics and AI to molecular Imaging for Prostate Cancer”.
We are hopeful you agree that this final revision will further update our comprehensive review. All the comments of “reviewer 1” have been addressed, as shown in this version of the manuscript, along with this point-by-point response to his comments.
All corresponding are red changes in the manuscript.
Reviewer #1:
-
General comment:
“Thanks for improving the manuscript according to my suggestions.”
Response:
We appreciate the opportunity to consider our manuscript for publication.
-
Specific comments:
“however, some other points warrant mention:”
- all the added information misses of related citations.
Response:
Thank you for your comment.
We have added citations 3, 5, 6, 122 and 123.
- In lines 47-51, do the authors think that AI could be helpful "only" to oncologists in predicting PCa and outcomes?
Response:
This is not what was stated or implied in the manuscript. We apologies for any misunderstanding.
- imaging does not diagnose PCa. The diagnosis is made by urologists by biopsies.
Response:
We completely agree; we have rephrased and clarified, appropriately (lines 53-62 in the revised manuscript).
- I don't think that sentences like ". See for instance Delgadillo et al. [16] in relation to radiotherapy, Ferro et al. [17], Penzkofer et al. [18] and Sun et al. [19]. For this reason, this review will focus on radiomics in molecular imaging in PET and in CT." should be included in a scientific manuscript.
Response:
We completely agree and apologize for this expression. Our intention was to explain what the review does not cover as well as what it does cover. We have now deleted these sentences (lines 156-157 in the revised manuscript).
- with "Ga" do the authors refer to Gallium?
Response:
Thank you; we have added in the revised version of the manuscript the relevant abbreviation (line 87 in the revised manuscript).
- doesn't seem polite to use "Introduction" and "Conclusions" as titles of subsections.
Response:
Thank you for your comment.
I would like to have the opinion of the assistant editor at that point, as I believe that we have just followed the styling of the journal – of note that I have published 37 manuscripts in MDPI journals.
- the entire manuscript is not well-organized. As mentioned in the previous review step, some sections are too long and some sentences are useless. thus the manuscript does not provide a high-quality overall overview of the topic, useful to the readers.
Response:
Thank you for your comment.
However, we have been congratulated by the reviewer 2 and reviewer 3 for a high-quality overall overview and this can be confirmed by the assistant editor. Their comments were as follows:
Reviewer 2: “The work provides a good overview of AI techniques for prostate cancer molecular imaging, especially PET/CT. The manuscript is well organized, moving from an introduction to PSMA imaging and systematically reviewing major AI methods and application areas. A comprehensive set of relevant references are cited. Both methodology papers and application studies are covered. The pros and cons of different techniques are discussed. Challenges and future directions are summarized well.
Overall, this well-written review comprehensively summarizes the state-of-the-art in this emerging application area for AI in medical imaging. The work offers a significant contribution as a reference for prostate cancer imaging and AI researchers.”.
Reviewer 3: “This review by William Tapper and co-authors provides an overview of AI applied to PET/CT in PCa. Overall, the review is well-executed.”
Regardless, we are grateful for your comment and we have tried to modify some long sections, as you kindly suggested.
-
Comments on the Quality of English Language
Improvements are still needed.
Response:
We have further improved the quality of English language; thank you.
Please note that the authors are all fluent English speakers. For Tapper, Thomas and Evans English is their first language. Carneiro has worked in Australia and the UK for many years. Mikropoulos and Boussios are medically qualified and have worked in UK hospitals for many years. In addition the paper was reviewed before submission by two native English speakers working at the UK National Physical Laboratory.